# An Overview on the Histogenesis and Morphogenesis of Salivary Gland Neoplasms and Evolving Diagnostic Approaches

**Janaki Iyer** [1] **, Arvind Hariharan** [1] **, Uyen Minh Nha Cao** [1,2] **, Crystal To Tam Mai** [1] **, Athena Wang** [1] **, Parisa Khayambashi** [1] **, Bich Hong Nguyen** [3] **, Lydia Safi** [1] **and Simon D. Tran** [1,*]

1 McGill Craniofacial Tissue Engineering and Stem Cells Laboratory, Faculty of Dentistry, McGill University, 3640 University Street, Montreal, QC H3A 0C7, Canada; janaki.iyer@mail.mcgill.ca (J.I.); arvind.hariharan@mail.mcgill.ca (A.H.); cmnuyen.chrhm18@ump.edu.vn (U.M.N.C.); crystal.mai@mail.utoronto.ca (C.T.T.M.); athw13@student.ubc.ca (A.W.); parisa.khayambashi@mail.mcgill.ca (P.K.); lydia.safi@mail.mcgill.ca (L.S.)
2 Department of Orthodontics, Faculty of Dentistry, Ho Chi Minh University of Medicine and Pharmacy, Ho Chi Minh City 700000, Vietnam
3 CHU Sainte Justine Hospital, Montreal, QC H3T 1C5, Canada; bich.hong.nguyen@umontreal.ca
* Correspondence: simon.tran@mcgill.ca

**Simple Summary:** Diagnosing salivary gland neoplasms (SGN) remain a challenge, given their underlying biological nature and overlapping features. Evolving techniques in molecular pathology have uncovered genetic mutations resulting in these tumors. This review delves into the molecular etiopatho-genesis of SGN, highlighting advanced diagnostic protocols that may facilitate the identification and therapy of a variety of SGN.

**Abstract:** Salivary gland neoplasms (SGN) remain a diagnostic dilemma due to their heterogenic complex behavior. Their diverse histomorphological appearance is attributed to the underlying cellular mechanisms and differentiation into various histopathological subtypes with overlapping fea-tures. Diagnostic tools such as fine needle aspiration biopsy, computerized tomography, magnetic resonance imaging, and positron emission tomography help evaluate the structure and assess the staging of SGN. Advances in molecular pathology have uncovered genetic patterns and oncogenes by immunohistochemistry, fluorescent in situ hybridization, and next–generation sequencing, that may potentially contribute to innovating diagnostic approaches in identifying various SGN. Surgical resection is the principal treatment for most SGN. Other modalities such as radiotherapy, chemother-apy, targeted therapy (agents like tyrosine kinase inhibitors, monoclonal antibodies, and proteasome inhibitors), and potential hormone therapy may be applied, depending on the clinical behaviors, histopathologic grading, tumor stage and location, and the extent of tissue invasion. This review delves into the molecular pathways of salivary gland tumorigenesis, highlighting recent diagnostic protocols that may facilitate the identification and management of SGN.

**Keywords:** salivary glands; salivary gland neoplasms; epithelial tumors; head and neck cancer; molecular pathology; diagnostic advances

## 1. Introduction

Salivary glands are tubulo-acinar exocrine organs that embryonically initiate in the sixth–eighth week of intrauterine life. The parotid gland is believed to arise from the oral ectoderm, while the submandibular and sublingual glands are from the embryonic endoderm [1,2]. Their development is attributed to the physiologic process of 'branching morphogenesis', described as the rearrangement of a single epithelial bud to generate multiple acinar and ductal units, through continuous multi-directional branching [3]. 'Epithelial–mesenchymal interaction', described as a secondary induction of the epithelium

by its underlying mesenchyme, is also essential for the normal development of salivary glands [4]. This cascade of events ultimately forms multiple secretory units, each consisting of a terminal acinar (serous/mucous) cell, myoepithelial cell, intercalated duct, striated duct, and excretory duct, as elaborated in Figure 1 [1,2].

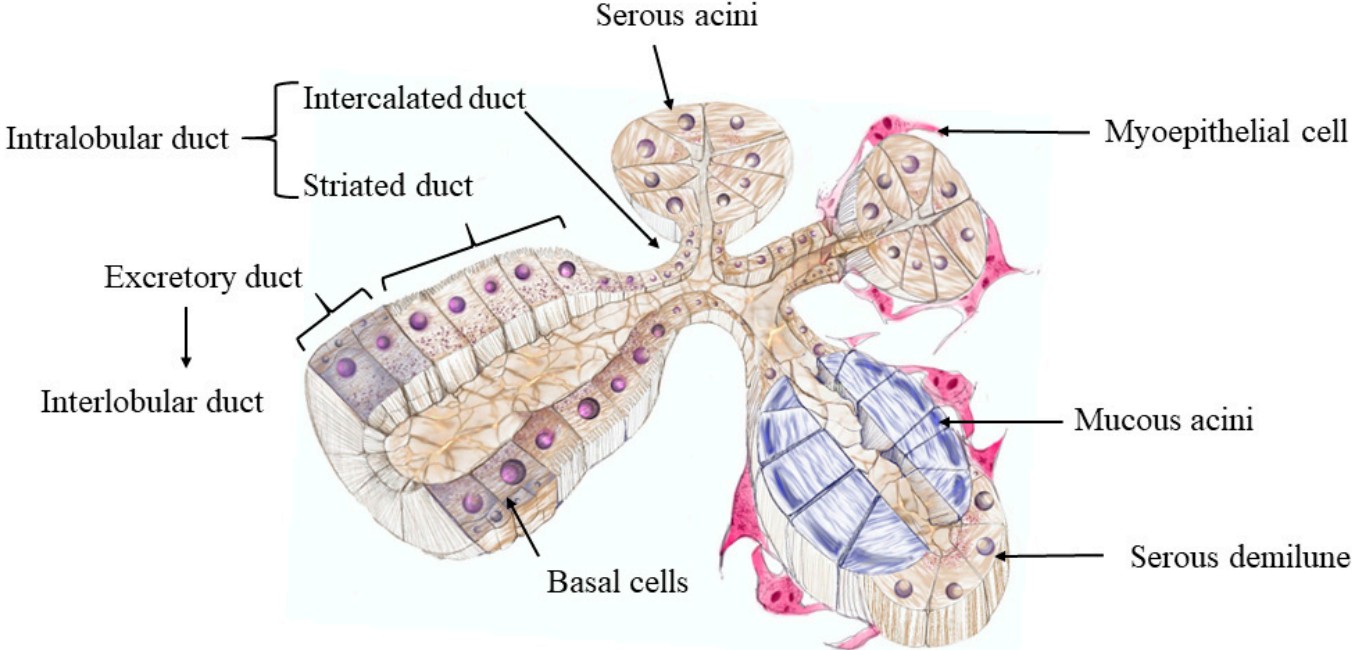

**Figure 1.** Schematic representation of the histology of salivary glands. Reprinted from [5,6] with permission. Ducto-acinar architecture of salivary glands is divided by capsular connective tissue septa into lobules. Each lobule consists of numerous serous/mucous/mixed (also known as serous demilune around mucous acini) secretory acini that are enveloped by myoepithelial cells. The secretory acini unite to form intercalated ducts (lined by simple squamous to low cuboidal epithelium and wrapped by myoepithelial cells). Saliva is secreted by acinar cells and drains into the striated ducts that are lined by simple or pseudostratified columnar epithelial cells. This ductal lining transforms into stratified squamous epithelium supported by basal cells. Ultimately, the striated ducts drain into the excretory ducts (also known as interlobular ducts) with tall columnar epithelial cells. Figure adapted from Tran et al., 2019 and Proctor and Shaalan, 2018 [5,6].

The architecture of salivary glands is two-tiered, consisting of luminal (acinar and ductal) and abluminal (myoepithelial and basal) cells [1,2]. These cells enter the cell cycle rapidly, thus acting as potential targets for neoplastic transformation. The estimated global incidence of salivary gland neoplasms (SGN) ranges from 0.4 to 13.5 cases per 100,000 annually, and constitutes approximately 3 to 6% of head and neck tumors [7]. The diverse histomorphological appearance of SGN is attributed to their heterogenic and complex cellular behavior. Differentiation into various histopathological subtypes with overlapping features both within tumors and in different regions of the same tumor, results in significant diagnostic challenge [1,2,8]. Advances in molecular pathology have uncovered genetic patterns and biomarkers that may potentially contribute to innovating diagnostic approaches in identifying various salivary gland pathologies [9]. In this paper, we delve into the molecular pathways of salivary gland tumorigenesis, highlighting recent diagnostic protocols that may facilitate the identification and management of SGN.

### 1.1. Pathogenesis of Salivary Gland Neoplasms (SGN)

1.1.1. Histogenic Concepts

Former concepts of pathogenesis of SGN were focused on the histologic cell of origin. The adult salivary glands consist of reserve cells that are believed to replicate pathologically

to form SGN. The four commonly hypothesized histogenetic theories, as depicted in Figure 2A, are as follows:

- Basal reserve cell or progenitor cell theory: This concept is based on the assumption that basal cells of the excretory and intercalated ducts function as reserve cells for more highly differentiated components of the functional salivary complex [1,2].
- Pluripotent unicellular reserve cell theory: Evolution of the basal reserve cell theory stated that the basal cells of excretory ducts were responsible for the development of all salivary gland units [1,2];
- Semi-pluripotent bicellular reserve cell theory: A more plausible interpretation of the reserve cell theory suggested that the basal cells of the excretory duct (excretory duct reserve cells) produced squamous or mucin-producing columnar cells, and those from the intercalated ducts (intercalated duct reserve cells) were responsible for development of intercalated, striated, and acinar elements [1,2,10];
- Multicellular theory: Further investigation provided evidence that all mature cell types, including acinar and basal cells in salivary gland tissue were capable of proliferation. This theory presumes that SGN originated from the differentiated or adult cell counterpart from within the functional salivary ducto-acinar complex [1,2].

### 1.1.2. Morphogenic Concepts

Apart from the cell of origin, a pathologist typically considers the differentiation process and arrangement of tumor cells as crucial when classifying the neoplasm. In order to overcome challenges in determining the cell of origin, the morphogenic approach of cellular differentiation facilitates immunohistochemical and ultrastructural analyses, leading to a more accurate diagnosis. The bicellular differentiation in the development of salivary glands can be revisited in the pathogenesis of SGN, along the ducto-acinar complex. At each level of the salivary gland, cellular differentiation may result in different models of tumor cell subtypes, as shown in Figure 2B. The synthesis of extracellular matrix (ECM) by the basal lamina and its position between the cellular compartments affects the histomorphology and eventually the classification and diagnosis of the neoplasm. This highlights the need for the morphogenic theory [1,2,11–13].

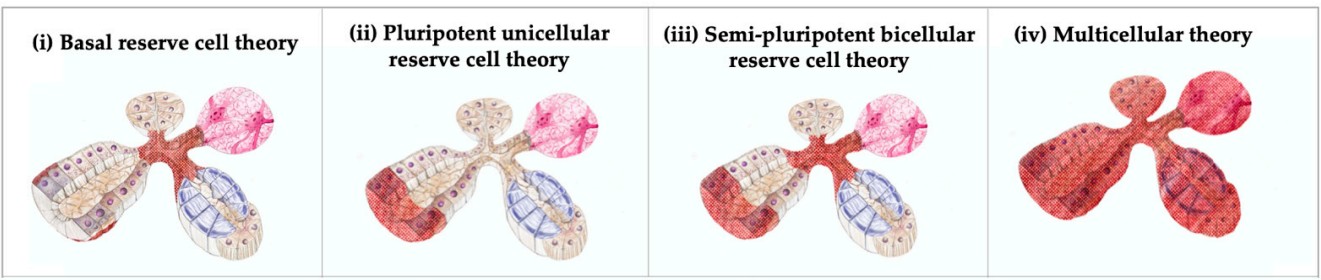

(**A**) Histogenic concepts.

**Figure 2.** *Cont.*

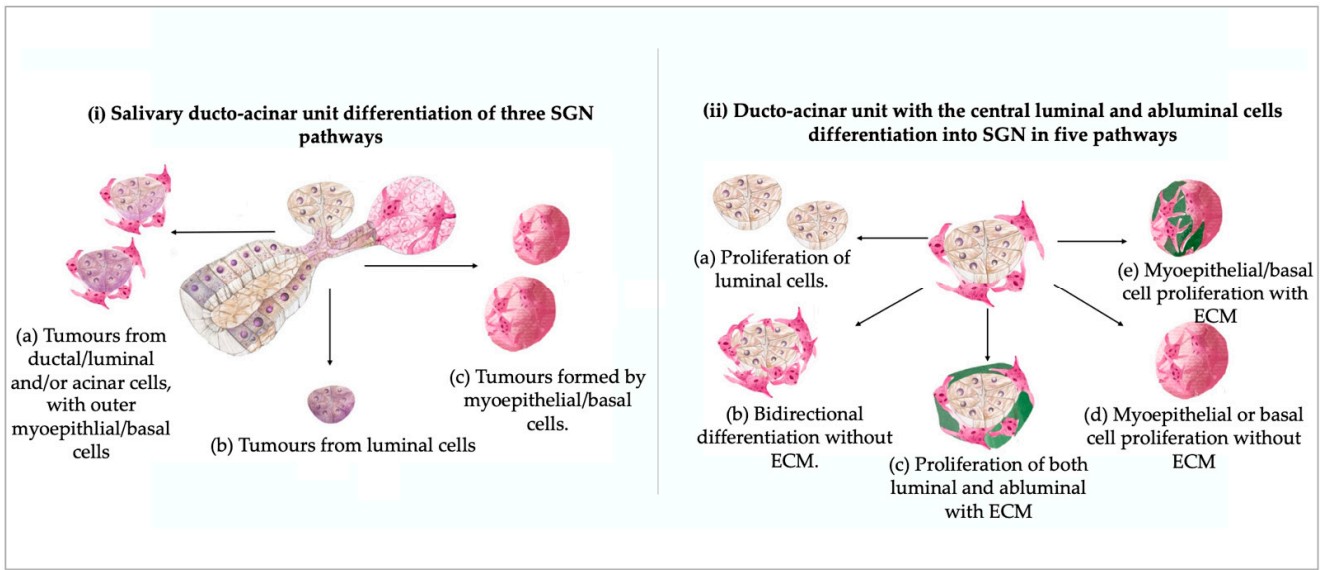

(**B**) Morphogenic concepts.

**Figure 2.** Schematic representation of the pathogenesis of SGN. (**A**) Histogenic concepts. (**B**) Morphogenic concepts. Reprinted from [2,5,6,13] with permission. Figure 2A depicts the four histogenic concepts that emphasize reserve cells of the salivary gland that replicate pathologically to form SGN (the cell type is highlighted in red for each theory): (i) Basal reserve cell theory–basal cells of excretory and intercalated ducts; (ii) pluripotent unicellular reserve cell theory–basal cells of the excretory duct; (iii) semi-pluripotent bicellular reserve cell theory–basal cells of the excretory duct and intercalated duct; (iv) multicellular theory–all mature acinar and basal cells. Figure 2B highlights the morphogenic concept that emphasizes the differentiation process and arrangement of tumor cells in SGN development: (i) Salivary ducto-acinar unit showing potential for differentiation of three SGN pathways: (a) tumors arising from combination of ductal/luminal and/or acinar cells, with outer myoepithelial/basal cells; (b) tumors mainly originating from luminal cells that may differentiate into non-specific ductal, acinar or goblet cells, and/or combination of these cells; (c) tumors almost entirely formed by myoepithelial/basal cells; or (ii) cross-section of the ducto-acinar unit with the central luminal (luminal/acinar) and surrounding abluminal cells (myoepithelial and basal cells) which can further differentiate into SGN: (a) proliferation of luminal cells; (b) bidirectional differentiation without extracellular matrix (ECM) materials; (c) proliferation of both luminal and abluminal cells with foci of extracellular matrix materials; (d) myoepithelial or basal cell proliferation without extracellular matrix materials; (e) myoepithelial or basal cell proliferation with extracellular matrix materials. The figure was adapted from Sreeja et al. 2014 and Jagdish 2014 [2,13].

Dardick deemed cellular morphology and cellular differentiation, derived from differential gene expression of a stem cell, in conjunction with tumor ECM production, to be better predictors of SGN, when compared to a specific proposed cell of origin [14].

## 2. Classification of SGN

The classification of SGN is an ever-evolving process, given their varied histomorphological appearances, lack of uniformity, overlapping features, and diverse nature of individual entities. The transitional nature of SGN thus contributes to this diagnostic dilemma. The most recent and widely accepted classification of SGN is featured in the fourth edition of the World Health Organization (WHO) classification of head and neck tumors, as elaborated in Table 1 [15]. The current classification has a modified list, with inclusion and exclusion of several histopathological entities, as compared to the third WHO edition [16,17]. As the understanding of the biological behavior of lesions progresses both on a genetic and molecular level, newer variants of pre-existing neoplasms continue to emerge. Evolving immunohistochemical markers and innovative diagnostic approaches may result in a more accurate identification, thus offering effective treatments of SGN.

**Table 1.** WHO classification of SGN (2017). Reprinted from [15] with permission.

| Histopathological Variant | ICD-O Code | Histopathological Variant | ICD-O Code |
|---|---|---|---|
| Malignant epithelial tumors | | Benign tumors | |
| Acinic cell carcinoma | 8550/3 | Pleomorphic adenoma | 8940/0 |
| Secretory carcinoma | 8502/3 | Myoepithelioma | 8982/0 |
| Mucoepidermoid carcinoma | 8430/3 | Basal cell adenoma | 8147/0 |
| Adenoid cystic carcinoma | 8200/3 | Warthin tumor | 8561/0 |
| Polymorphous adenocarcinoma | 8525/3 | Oncocytoma | 8290/0 |
| Epithelial–myoepithelial carcinoma | 8562/3 | Lymphoadenoma | 8563/0 |
| Clear cell carcinoma | 8310/3 | Cystadenoma | 8440/0 |
| Basal cell adenocarcinoma | 8147/3 | Sialadenoma papilliferum | 8406/0 |
| Sebaceous adenocarcinoma | 8410/3 | Ductal papillomas | 8503/0 |
| Intraductal carcinoma | 8500/2 | Sebaceous adenoma | 8410/0 |
| Cystadenocarcinoma | 8440/3 | Canalicular adenoma and other ductal adenomas | 8149/0 |
| Adenocarcinoma, NOS | 8140/3 | | |
| Salivary duct carcinoma | 8500/3 | Other epithelial lesions | |
| Myoepithelial carcinoma | 8982/3 | Sclerosing polycystic adenosis | |
| Carcinoma ex pleomorphic adenoma | 8941/3 | Nodular oncocytic hyperplasia | |
| Carcinosarcoma | 8980/3 | Lymphoepithelial lesions | |
| Poorly differentiated carcinoma: | | Intercalated duct hyperplasia | |
| Neuroendocrine and non-endocrine | | | |
| Undifferentiated carcinoma | 8020/3 | Soft tissue tumors | |
| Large cell neuroendocrine carcinoma | 8013/3 | Hemangioma | 9120/0 |
| Small cell neuroendocrine carcinoma | 8041/3 | Lipoma/sialolipoma | 8850/0 |
| Lymphoepithelial carcinoma | 8082/3 | Nodular fasciitis | 8828/0 |
| Squamous cell carcinoma | 8070/3 | | |
| Oncocytic carcinoma | 8290/3 | Hematolymphoid tumors | |
| Borderline tumour | | Extranodal marginal zone lymphoma of MALT | 9699/3 |
| Sialoblastoma | 8974/1 | | |

The morphology codes are from the International Classification of Diseases for Oncology (ICD-O) (742A). Behavior is coded: 0 for benign tumours; 1 for unspecified, borderline, or uncertain behavior; 2 for carcinoma in situ and grade III intraepithelial neoplasia; and 3 for malignant tumors. The classification is modified from the previous WHO classification, taking into account changes in our understanding of these lesions. These new codes were approved by the IARC/WHO Committee for ICD-O. *Italics:* Provisional tumour entities. Grading according to the 2013 *WHO Classification of Tumours of Soft Tissue and Bone.* Information obtained from [15].

## 3. Diagnostic Workup and Recent Advances in Diagnosis

SGN often present with an enlarged mass, requiring further investigation for proper diagnosis. Fine needle aspiration biopsy (FNAB) has been commonly used to diagnose SGN [18]. However, due to the heterogeneity of the neoplasms, imaging procedures such as ultrasound (US), computerized tomography (CT), and magnetic resonance imaging (MRI) are commonly used to evaluate the structure and assess the staging of SGN [18,19]. While CT and MRI visualize structural changes, positron emission tomography (PET) visualizes any molecular changes [20,21]. Despite the variety of techniques involved in the evaluation of SGN, different cases may require specific imaging techniques for accurate diagnosis. Additionally, oncogenes are an innovative technique that can improve tumor classification.

### 3.1. Clinical History

SGN may be found in the parotid, submandibular, sublingual glands, accessory glands, and minor salivary glands, and most are initially identified by the swelling of these glands [22]. Furthermore, symptoms that suggest malignancy include pain, rapid tissue growth, or loss of nerve function [23,24]. In clinical practice, it has been reported that minor SGN account for less than 25% of SGN [25], and smaller salivary glands have a higher incidence of malignancy. While only 20% of SGN are malignant [26], it is crucial to accurately differentiate benign from malignant neoplasms in order to devise an appropriate treatment plan.

### 3.2. Fine-Needle Aspiration Biopsy (FNAB)

FNAB is one of the first line procedures used to diagnose SGN on account of its easy, inexpensive, highly accurate, quick, and minimally invasive nature [27]. This technique entails using a fine gauge needle to collect cells. After alcohol fixation and drying, the cellular aspirate is stained with Papanicolaou stain and can be immediately evaluated and diagnosed [27]. FNAB results are universally reported using the Milan's system, as seen in Table 2 [28]. Edizer et al. (2016), evaluated the ability of FNAB to differentially diagnose salivary gland masses by comparing the preoperative FNAB results with the postoperative definitive histopathological results of 285 patients. Their FNAB results were 92.6% accurate compared to the definitive histopathological results. This demonstrated that FNAB is useful in benign and malignant tumor differentiation. However, they do have some limitations, which involve relatively high non-diagnostic results, possibly due to bleeding, low cellularity, necrosis, or erroneous technique [27]. In addition, some potential outcomes in the final histopathological examination include squamous metaplasia and fibrosis. However, these do not interfere with the definitive diagnosis [27].

**Table 2.** Milan's system of FNAB reporting for SGN [28].

| Diagnostic Category | Risk of Malignancy % | Management |
|---|---|---|
| Non-diagnostic | 25 | Clinical and radiologic correlation/repeat FNAC |
| Non-neoplastic | 10 | Clinical follow-up and radiological correlation |
| Atypia of undetermined significance (AUS) | 20 | Repeat FNAC or surgery |
| Neoplasm: benign | <5 | Surgery or clinical follow-up |
| Neoplasm: salivary gland neoplasm of uncertain malignant potential (SUMP) | 35 | Surgery |
| Suspicious for malignancy (SM) | 60 | Surgery |
| Malignant | 90 | Surgery |

Information obtained from [28].

### 3.3. Ultrasound (US)

US is a highly effective non-invasive technology that can be used in the differential diagnosis of SGN [29]. The technology uses high-frequency sound (ultrasonic) waves to generate images of internal tissues and organs [30]. Modern USs have demonstrated greater success in providing precise measurements, localization, and evaluation of the structures of various SGN, as highlighted in Table 3 [29,31,32]. In a study conducted by Bialek et al. (2003), the role of the US in the differentiation and diagnosis of Pleomorphic Adenomas (PA) was analyzed. By using a modern US machine, in conjunction with high-resolution probes and tissue harmonic imaging, they were able to detect 96% of malignant salivary glands in patients with solid lesions. Modern USs are considered highly valuable, dependable, and useful in the differential diagnosis of SGN; however, they possess some

limitations [29]. USs are unable to properly assess lesions located in obscure areas (i.e., deep lobe of the parotid gland, behind bones) and are inadequate in differentially diagnosing small lesions [29].

### 3.4. Computerized Tomography (CT)

In conjunction with the US, contrast-enhanced computerized tomography (CT) is an imaging technique often used to obtain a more detailed view of deeper masses (Table 3) [31–34]. Given that CT scanning exposes patients to high levels of radiation, variations of CT such as cone beam CT (CBCT) have been used as an alternative measure since it emits relatively decreased levels of radiation [35]. Furthermore, in a study by Jung et al. (2020), researchers found that single-phase CT scanning may be a low-radiation alternative in the differentiation of tumors [18]. They compared the texture analysis parameters in single-phase CT and conventional two-phase CT, to differentiate between two common types of benign tumors: Warthin tumor (WT) and PA. The authors found that the differential parameters between WT and PA from a single-phase CT were similar to those of a two-phase scan. Moreover, they found that the patient was exposed to less radiation during texture analysis via the single-phase imaging. Thus, researchers concluded that this tool could be a minimally invasive method in the investigation of benign SGN [18]. However, further testing is required to assess whether these findings may be extended to the differentiation of malignant neoplasms.

### 3.5. Magnetic Resonance Imaging (MRI)

Despite MRI being relatively more costly and requiring more time to produce images [36], its major benefit is that it is free of radiation. Additionally, researchers have previously deemed MRI to be the most suitable imaging technique in the assessment of parotid gland tumors and relation to adjacent vital structures as it offers a high contrast resolution of the soft tissues (Table 3) [31,32,37]. Parameters investigating malignancy include tumor border configuration, invasion of adjacent tissues, T1- and T2- weighted signal intensity, and time–intensity curve with constant enhancement [38,39]. Common MRI findings that favor malignancy include low T2 signal, heterogenous enhancement, lesional growth with ill-defined, or blurry tumor borders that may invade into the adjacent structures and lymph nodes. Malignant SGN imaging may reveal cystic changes, central necrosis, perineural infiltration, accompanied by regional or distant metastasis [40]. Low-grade malignant tumors may resemble benign lesions; however, the difference of contents of the cystic component of benign lesions may be revealed as increased hyperintense T1-weighted images [40,41]. Although MRI is the preferred imaging technique for SGN, this diagnostic approach cannot be employed among patients allergic to contrast dyes. Therefore, in a study conducted by Takumi et al. (2021), they investigated a combination of non-contrast MRI techniques to enhance the diagnostic performance in differentiating between benign and malignant SGN [36]. They focused on three non-contrast MRI parameters: apparent diffusion coefficient, tumor blood flow, and amide proton transfer related signal intensity. Upon studying each parameter individually, diagnostic performance was found to be limited. However, when these parameters were combined, there was a significant increase in the accuracy of the diagnosis, thus leading the authors to conclude that this multiparametric approach of using non-contrast MRI may improve the differentiation of the nature of the SGN [36].

### 3.6. Positron Emission Tomography (PET)

PET is a non-invasive imaging technology that uses radioactive tracers to visualize and evaluate tissues and organs for the presence of diseases, including cancer [42]. Once these tracers are intravenously injected, they gather in areas of higher chemical activity, often indicating areas of disease [42]. These tracers emit radiation that can be detected by the PET scanner, which generates an image map for assessment [42]. Roh et al. (2007), evaluated the role of PET, using 18F-fluorodeoxyglucose (FDG) as tracers among patients with salivary gland cancers (Table 3) [43]. They were able to detect 91.2% of primary

tumors in patients and concluded that 18FDG-PET is clinically useful in histologic grading and initial staging of salivary gland malignancies. However, the technology does have some limitations. Occasionally, normal physiologic uptake of radioactive tracers occurs, which often mimics or hides existing neoplasms [43]. Additionally, low-grade malignancies frequently have lower tracer uptake than high-grade malignancies [43]. Therefore, these limitations may lead to undetected SGN [43].

**Table 3.** Summary of diagnosing techniques and parameters [31,32,43].

| Imaging Technique | Principle | Interpretation Guidelines (Parameters Studied) | Sensitivity | Specificity |
|---|---|---|---|---|
| Ultrasound (US) [31,32] | Use of high-frequency sound waves to generate images of internal tissues and organs for diagnosis | Tumor: location, dimensions, shape, structure, margins, vascularization | 63% | 92% |
| Computerized tomography (CT) [31,32] | Using a series of X-ray images to produce a cross-sectional image of tissues for diagnosis | Tumor boundary, enhancement pattern, calcification | 83% | 85% |
| Magnetic resonance imaging (MRI) [31,32] | Use of magnetic field and radio waves to produce images for diagnosis | T1-, T2-weighted images for tumor localization, extent, perineural infiltration and relation to adjacent structures. Other parameters: apparent diffusion coefficient, time–intensity curve, amide proton transfer-telated signal intensity | 81% | 89% |
| Positron emission tomography (PET) [43] | Use of radioactive tracers to visualize and evaluate tissues and organs for diagnosis | Tumor maximum standardized uptake value, clinicopathlogic parameters (local tumor invasion, T and N categories, TNM stage, loco-regional and distant lymph node metastasis) | 80.5% (cervical lymphnode levels with metastases) | 89.5% (cervical lymphnode levels with metastases) |

Information obtained from [31,32,43].

### 3.7. Biopsy and Histopathological Diagnosis

Following the procurement of cells through biopsy, histopathological diagnosis assesses the SGN morphologically [44]. Different types of SGN can be identified based on their location and cell composition. For instance, acinic cell carcinoma is usually present in the parotid gland with its cells composed of acinar and intercalated types [44]. Meanwhile, mucoepidermoid carcinoma is found in major or minor salivary glands. These tumors are composed of squamoid, mucous, and intermediate cells, and may also contain solid or cystic regions [44]. Salivary duct carcinoma is mainly found de novo, or derived malignantly from carcinoma-ex-pleomorphic adenoma in the parotid gland. These neoplasms are characterized by locoregional metastasis [44]. Moreover, the most common benign and malignant SGN found at the parotid and submandibular glands were pleomorphic adenoma and adenoid cystic carcinoma, respectively [45]. Thus, accounting for these histopathological features can assist in the diagnosis of SGN.

### 4. Oncogenes as a Novel Diagnostic Tool

Although SGN can be diagnosed with a variety of the aforementioned methods, given their transient nature, they may still pose as a diagnostic challenge for clinicians. The characteristics of many SGN tend to overlap, particularly histopathologically. Recent progress has been made with novel diagnostic tools to identify the genetic changes that occur in SGN. This has led to a more accurate diagnosis, resulting in more effective

treatments and, therefore, better prognosis [9,46]. All SGN have certain genetic alterations that can be categorized, as in Table 4, according to their role in diagnosis, prediction, and prognosis. [47].

**Table 4.** Diagnostic, predictive, and prognostic markers in SGN. Reprinted from [47] with permission.

| Tumor Subtype | Genetic/Molecular Alterations | Role of Alteration |
|---|---|---|
| Pleomorphic adenoma | *PLAG1* alterations | Diagnostic |
| | *HMGA2* alterations | Diagnostic |
| | *HER2* overexpression | Predictive for therapeutic response |
| | *AR* overexpression | Predictive for therapeutic response |
| Mucoepidermoid carcinoma | *CRTC1–MAML2* fusion | Diagnostic/prognostic |
| | *CRTC3–MAML2* fusion | Diagnostic/prognostic |
| Adenoid cystic carcinoma | *MYB/MYBL1* rearrangements | Diagnostic/predictive (*MYB* overexpression for therapeutic response) |
| | *MYB–NFIB* fusion | Diagnostic |
| | *NOTCH1* mutations | Prognostic |
| Acinic cell carcinoma | *NR4A3* rearrangements | Diagnostic |
| Polymorphous low-grade adenocarcinoma | *PRKD1/2/3* rearrangements | Diagnostic |
| | *PRKD1 E710D* hot spot mutations | Diagnostic/prognostic |
| Clear cell carcinoma | *EWSR1–ATR* fusion | Diagnostic |
| Salivary duct carcinoma | *AR* gene alterations | Diagnostic/predictive for androgen–deprivation therapy response |
| | *ERBB2* amplifications | Diagnostic/prognostic |
| | *TP53, PIK3CA, H-RAS* mutations | Diagnostic/prognostic (only *TP53*) |
| | *KIT, EGFR, BRAF, AKT1, N-RAS, FBXW7, ATM, NFI* mutations | |
| | Loss of heterozygosity of *CDKN2A, p16, PTEN* | Diagnostic |
| Myoepithelial carcinoma | *EWSR1* rearrangements | No confirmatory role |
| Epithelial–myoepithelial carcinoma | *HRAS* mutations | No confirmatory role |

Information obtained from [47].

While traditional diagnostic methods have been successful, a clearer understanding in the cellular and molecular mechanisms of SGN is necessary. The three novel diagnostic tools that have revolutionized the characterization of SGN are immunohistochemistry (IHC), fluorescent in situ hybridization (FISH), and next-generation sequencing (NGS) [9,48]. The discovery of the genetic alterations, their significance in oncogenesis of common SGN and the usage of these novel diagnostic tools are further analyzed in the following sections.

### 4.1. Pleomorphic Adenoma

Pleomorphic adenoma (PA) is the most common salivary gland tumor and is categorized as a mixed type of tumor due to the presence of epithelial and myoepithelial cells [49]. The incidence of PA is increasing due to the prolonged exposure to radiation during head and neck cancer treatment [47]. Due to its varying morphology, it is difficult to differentiate it from other tumors of the same origin.

There are several translocations that have been identified for PA. Genetic aberrations occur involving the transcription factor genes *PLAG1* and *HMGA2*. *PLAG1* is a proto–oncogene located on chromosome 8q12 [50]. Overexpression leads to the activation of various signaling pathways, including WNT or HRAS, which determine the fate of cells [47]. *HMGA2* is located on chromosome 12q14 and is the second most common genetic event occurring in PA. Though unclear, the molecular mechanism for its overexpression is likely to encode for an architectural transcriptional factor that binds to the adenosine–

thymine DNA sequences, thus acting as transcription regulators for cell death, growth, and proliferation [47].

*PLAG1* and *HMGA2* are important diagnostic markers and can be detected with IHC. The detection of *PLAG1* has been found to have clinicopathological impacts, and is supported by histopathological findings [50,51]. The overexpression of *PLAG1* by IHC has helped differentiate between PA and other SGN, such as adenoid cystic carcinoma (ACC), with high specificity [52]. A study by Mito et al. (2017) also showed the importance of IHC in the detection of *HMGA2.* They found that it is a highly specific marker for PA compared to other histologically mimicking tumors [53].

FISH is now at the forefront of SGN diagnosis due to the discovery of novel oncogenic fusions and gene translocations. It has been useful in diagnosing PA with the expression of fusions involving *PLAG1* and *HMGA2* [54]. Evrard et al. (2017) showed how the use of FISH facilitated salivary gland cytology and thus, the assessment of the extent of surgery. They concluded that the addition of FISH in the detection of *PLAG1* to conventional cytological analysis increased overall sensitivity and eliminated the need to use frozen sections for a diagnosis [55].

PA has the potential to transform into carcinoma ex pleomorphic adenoma (Ca ex–PA) adding to the diagnostic challenge. The expression of *PLAG1* and *HMGA2* is common for both tumors [47]. In a molecular study that used FISH to determine the similarities between PA and Ca ex–PA, the reviewed cases displayed evidence of metastasis. However, they appeared histologically benign which further complicates differentiation [56]. There is evidence that Ca ex–PA could be differentiated by overexpression of *TP53, AR*, and *HER2* genes. However, further research is required to confirm whether mutations of these genes could signify malignant transformation and hence be used as predictive biomarkers.

### 4.2. Mucoepidermoid Carcinoma

Mucoepidermoid carcinoma (MEC) is the most common malignancy of the salivary glands and can occur in both children and adults. It is characterized by the increased proliferation of the excretory cells [9]. While the etiology remains controversial, some studies have shown the implications of viruses [9]. The diagnostic and prognostic markers involve fusion proteins derived from chromosomal rearrangements [57].

The genetic aberrations involve *CRTC1–MAML2* or *CRTC3–MAML2* fusions, with the latter being more important [58,59]. *CRTC1* is located on chromosome 9 and it encodes protein from the CREB family to enhance transcription. The CREB protein is responsible for regulating all genes involved in proliferation and differentiation. The *MAML2* gene is located on chromosome 11 and it encodes for the nuclear proteins responsible for the activation of the NOTCH pathway, which is one of the most common signaling pathways activated during tumorigenesis [47].

While *CRTC1–MAML2* fusion is detected in most cases of MEC, the molecular mechanisms have yet to be clearly understood. Once fusion occurs, the protein activates the transcription of CREB target genes to contribute to tumorigenesis. A study by Chen et al. (2021) showed that *CRTC1–MAML2* fusion could be modulated as a therapeutic target. After its elimination in mice, MEC xenografts demonstrated no further growth [58]. Earlier studies expressed this fusion as a potential prognostic marker due to its tendency to indicate a favorable prognosis in young patients [47]. Recent studies, however, have disproved this theory due to increasingly strict MEC diagnostic guidelines, especially in early-stage MEC [60].

Expression of *MAML2* using FISH has been acclaimed to be very useful. It is a relatively straightforward diagnosis considering that the expression of *MAML2* is exclusive to MEC [52]. It is particularly useful in diagnosing the oncocytic variants of MEC. These variants are more problematic to diagnose since they mimic other SGN, such as acinic cell carcinoma (AciCC) [52]. Although NGS has improved diagnostic accuracy, the *MAML2,* FISH, may sometimes exhibit negative results, notably in the oncocytic variants of MEC [61]. Case studies have shown that whenever FISH has failed to express fusion, NGS has

validated its potential as a confirmatory test [61]. The prognostic role of *MAML2* can be seen using IHC since it is thought that the *CTRC1–MAML2* fusion is a downstream target of the EGFR ligand, amphiregulin (AREG). A study by Shinomiya et al. (2016) supported this finding, where the overexpression of AREG and EGFR was characterized by IHC in MEC samples, which played a role in tumor growth and survival [62].

### 4.3. Adenoid Cystic Carcinoma

Adenoid cystic carcinoma (ACC) is another common malignant tumor of the salivary glands. It is slow-growing and composed of epithelial and myoepithelial cells of different origins [9]. Given its high recurrence, it has a very poor prognosis due to its metastatic capability and associated perineural invasion. Current treatment protocols involve surgery, followed by post-operative radiotherapy (PORT), which have been fairly successful [63]. PORT has shown to be an effective adjuvant to surgery and to minimize the incidence of recurrence [64]. However, studies have shown that it does not really affect the overall survival rates of patients, therefore questioning its effectiveness [65]. For more effective diagnosis and management strategies, it is necessary to understand the underlying genome alterations when studying recurrent ACC tumors [47].

Studies found that recurrent ACC showed alterations in the NOTCH pathway when compared to primary ACC cases. These mutations in the NOTCH pathway are significant as they could lead to potential therapeutic targets [66]. Ferraroto et al. (2017) concluded that the NOTCH mutations were indicative of a more distinct form of ACC, exhibiting metastasis in bone and liver; however, this was minimized by NOTCH inhibitors [67].

The main genomic alteration that characterizes ACC is the *MYB–NFIB* gene fusion. Overexpression of *MYB* is a diagnostic characteristic feature of ACC. It is located on chromosome 6q and it encodes for a transcription factor that regulates cell proliferation and differentiation of hematopoietic, colonic, and neural progenitor cells [47]. *NFIB* is located on chromosome 9q and is also a key regulator for hematopoietic and epithelial cells. A study by Rettig et al. (2016) presented an overexpression of *NFIB* in ACC, suggestive of an alternative oncogenetic pathway [68]. Whole genome sequencing has also revealed enhanced translocation, leading to the overexpression of *MYB*. This provided another insight into the downstream process of *MYB* in different ACC lineages [69].

Overexpression of *MYB* is thought to impact DNA repair, apoptosis, cell migration, and cell signaling for cell cycle control [47]. Xu et al. (2019) inferred that salivary ACC tissue samples displayed a higher expression of *MYB* when compared to normal salivary tissue and was associated with metastatic potential [70]. Detecting *MYB–NFIB* fusion can be difficult using IHC since *MYB* overexpression is also seen in other SGN. Thus, this *MYB–NFIB* fusion is detected using FISH [44]. NGS has also been proven useful in diagnosing ACC. In a recent case study, the presence of a *MYB–NFIB* fusion was detected by NGS in a suspected case of ameloblastoma with histopathological variations [61]. This emphasizes the significance of introducing these novel diagnostic tools for a more accurate diagnosis.

### 4.4. Acinic Cell Carcinoma

Acinic cell carcinoma (AciCC) is a low-grade malignancy consisting of both ductal and acinar cells with the presence of basophilic cytoplasm. It is the third most common malignancy of the salivary glands and is slow progressing but can metastasize to local and distant sites [9]. Though the knowledge of its molecular aberrations remains limited, it is commonly characterized by the expression of *DOG1* (a membrane channel protein), while the prominent genetic aberration is the translocation of *SCPP–NR4A3* [47].

*SCPP* is a secretory phosphoprotein that contains several genes responsible for producing salivary contents, bone, dentin, and enamel. It is located on chromosome 4q13 [36]. *NR4A3* is an important nuclear receptor that is located on chromosome 9q31 and it encodes for the steroid–thyroid hormone–retinoid receptor [47,71]. The upregulation of *NR4A3* increases the expression of target genes and influences cell proliferation. Another rare genetic fusion that can be seen is *MSANTD3–HTN3* translocation, which is characteristic

in variants with a more serous nature [72]. *MSANTD3* encodes for a poorly characterized protein, whereas *HTN3* is exclusively present in the saliva and functions as an antimicrobial peptide [71].

Diagnosis is straightforward as the *SCPP–NR4A3* is exclusive to AciCC. Moreover, the immunoexpression of *DOG1* is also a characteristic finding for AciCC [47]. IHC has proven to be more specific than FISH for the expression of *NR4A3* and has been found to be a specific and sensitive novel marker [71]. IHC has also shown relatively high specificity for the expression of *MSANTD3*. However, further studies are required to validate its role as a diagnostic marker in AciCC.

### 4.5. Polymorphous Adenocarcinoma

Polymorphous Adenocarcinoma (PAC) is an epithelial tumor most commonly found in the minor salivary glands. It is a relatively rare tumor and is usually associated with a favorable prognosis [9]. It was previously named "polymorphous low-grade carcinoma," and was renamed by the WHO (2017) due to its aggressive nature [47]. While cribriform adenocarcinoma has recently been incorporated into the PAC group of SGN due to their similar characteristics, it remains highly controversial whether they should be referred to as separate entities [73].

PAC is characterized by the mutation of *PRKD1*, a protein–kinase gene located on chromosome 14. It encodes a protein kinase that is involved in cellular processes including migration and differentiation, due to the signaling of the MAP kinase, RAS, and other cell survival and adhesion pathways [47]. The *PRKD1 E710D* hot spot mutation is a useful ancillary diagnostic marker along with the *PRKD1* mutations to differentiate between other SGN [73]. Although diagnosis is most likely done by visualizing the morphology, IHC can play a small role in certain instances [44]. Sebastiao et al. (2019) used FISH to demonstrate the genetic alterations of *PRKD1* as a diagnostic marker with reasonable success, particularly to identify nodal metastasis [74]. While *PRKD1 E710D* mutations as a prognostic marker has yet to be thoroughly researched, studies have observed its correlation with a metastasis–free tumor [73].

### 4.6. Clear Cell Carcinoma

Clear cell carcinoma (CCC) is a low-grade salivary tumor found in minor salivary glands and is characterized by the presence of clear cells [47]. It is identified by the appearance of *EWSR1–ATF1* fusion, which is a major genetic aberration [47]. *EWSR1* is an "Ewing's sarcoma" gene and is a member of the TET family protein group, located on chromosome 22q12. It encodes an RNA-binding protein, which is involved in gene expression, cell signaling as well as RNA processing, transport, and function [75]. *ATF1* is a transcription factor located on chromosome 12 and is an element of the CREB family of proteins. Studies have shown that tumorigenesis could occur due to the aberrant activation of *ATF1* upon fusion with *EWSR1* [75].

FISH is a very useful tool for diagnosing CCC as it can detect the *EWSR1–ATF1* fusion. An early study by Shah et al. (2013) demonstrated that FISH had higher sensitivity in detecting rearrangements of *EWSR1* in hyalinizing CCC [76]. At times, it can be difficult to differentiate between hyalinizing and odontogenic forms of CCC, as well as minor forms of MEC since they all exhibit translocations involving *EWSR1* [44]. NGS may be an even more accurate tool to differentiate and specify the genetic alterations between forms of CCC [44].

### 4.7. Salivary Duct Carcinoma

Salivary duct carcinoma (SDC) is a high-grade malignant neoplasm that usually arises from the parotid gland and is one of the most aggressive SGN. It is usually associated with a poor prognosis and frequent metastasis [9,47]. It is normally characterized by the expression of *AR*, an androgen receptor, located on chromosome Xq11-12 [47]. Studies have shown that treatment with androgen deprivation therapy may be effective with SDC since the expression of *AR* is equally seen in tumors of the prostate gland and breasts [77].

However, SDCs are also associated with somatic mutations of many other genes, including *TP53, ERBB2, HRAS,* and *PTEN*. This could be beneficial for more therapeutic targets of the associated downstream signaling pathways, such as mTOR, PI3K, Akt, and MAP kinase, which are major oncogenic drivers [78]. Multiple mutations have also shown to interfere with androgen response therapy, further necessitating the development of treatment strategies [79].

IHC and FISH have been useful in the detection of *AR* expression. *AR* immunoexpression has proven effective as a diagnostic and predictive biomarker which can further be treated with androgen deprivation therapies [77]. IHC detection of *TP53* and *ERBB2* mutations has been associated with a poor prognosis due to the activation of signaling pathways [47]. NGS has established insights on new fusions involving *ETV6–NTRK3* which allows for the possibility of new variants of SDC with different therapeutic targets [61].

### 4.8. Myoepithelial Carcinoma

Myoepithelial carcinoma is a rare SGN, consisting mainly of myoepithelial cells. It is characterized by *EWSR1* gene aberrations, making it difficult to distinguish from CCC [47]. However, FISH has helped with this distinction by demonstrating no evidence of fusion involving *EWSR1,* as seen in the case of CCC [47]. While it is considered to be chemoresistant, a study by Shenoy (2020) demonstrated that there is evidence of fusion between *EWSR1* and *POU5F1,* a feature in tumors arising from visceral organs [80]. He stated that it can be treated with combination chemotherapy in the treatment of Ewing's sarcoma [80].

### 4.9. Epithelial–Myoepithelial Carcinoma

Epithelial-myoepithelial carcinoma is a rare, bi-phasic tumor with a very low malignant potential and mainly characterized by *HRAS* mutations. However, other mutations involving *PIK3CA, CTNNB1,* and *AKT1* have also been reported to occur alongside *HRAS* mutations [47]. There is no concrete information regarding the molecular profile of this condition, and the extent to which these mutations can be used as diagnostic, prognostic, or predictive markers is unknown. However, in some studies, *HRAS* mutations have been seen as a diagnostic feature to differentiate it from other SGN mimickers. Further research is required due to the varied histology [81].

The recent advances in molecular pathology have aided deeper understanding of the etiopathogenesis of SGN. These varied histological subtypes also result in the need for tailored treatment options to optimize prognosis. Translational medicine, novel diagnostic tools, and improved technology promote newer and efficient therapeutic strategies for SGN.

## 5. The Management of SGN

SGN are abnormal tissue growths in the parotid, sublingual, submandibular, and minor salivary glands. The neoplastic conditions in the salivary glands present a wide variety of histological and clinical manifestations, ranging from benign to malignant and aggressive cancers [26]. Although surgical resection is the principal treatment for most SGN, the management of these tumors may vary depending on the clinical behaviors, histopathologic grading, tumor stage and location, and the extent of tissue invasion [82–84]. Thus, a thorough diagnostic and management plan should be made preoperatively [84].

### 5.1. Surgery

For noncancerous tumors, total surgical excision with a negative margin remains the standard treatment [84]. Enucleation is not a recommended option for benign tumors, as this technique may lead to higher incidences of recurrence and adjacent nerve damages [85]. Irradiation treatment exclusively has rarely been deemed as an effective treatment for SGNs. Moreover, postoperative radiotherapy is also inadvisable for benign tumors due to the associated risks of morbidity outweighing local benefits. However, in recurring cases, adjuvant radiotherapy has been proven to enhance locoregional control and reduce facial nerve damage [85]. With malignant neoplasms, the medical intervention often depends on

the stages of the tumor. When a tumor is in stage 1 (T1) or stage 2 (T2) without any evidence of nodal invasion, complete removal of the cancerous mass with optimal preservation of facial nerves is advisable. Long-term follow-up is crucial to prevent recurrences. In stage 3 (T3) and stage 4a (T4a), the primary tumors are greater than 4 cm and often infiltrate adjacent anatomical structures, resulting in bone invasion and/or perineural spread. At this stage, radical surgical resection of the tumors with any involved tissues should be performed [84]. For parotid gland tumors, a partial or total parotidectomy is often achieved at the advanced stage, and if there is any intraoperative evidence of peri-neural and connective tissue infiltration, the damaged tissues are also profoundly excised [83,86]. In more severe cases, lateral temporal bone or pharyngo-maxillary space resection would also be required [87]. SGN in submandibular and sublingual areas would need en-bloc resection of the tumors and related structures such as branches of facial nerves, the floor of the mouth, and a part of the mandible. A lymphadenectomy would also be crucial for the complete elimination of gross disease [83]. Selective neck dissection should be carefully evaluated even in confirmed cases of clinical N0 lymph node invasion. In the cases of clinical N+ neck invasion, a modified radical or total radical neck dissection is often performed to ensure the total removal of cancerous entities [84,88]. In stage 4b (T4b), the primary tumors become so extensive that they involve the craniofacial base and pterygoid plates. At this stage, total removal of the tumors may not be possible considering the risks of morbidity and the inability to achieve microscopically negative margins. In these inoperative cases, definite radiotherapy or a combination of chemotherapy and radiation would be implemented [84].

### 5.2. Radiotherapy

While surgical intervention with negative margins alone may be sufficient to terminate benign or small low-grade salivary gland tumors, malignant neoplasms would require adjuvant radiotherapy postoperatively. The application of adjuvant radiotherapy is often prescribed to patients in the advanced or recurrent stages, with lymph node metastasis, tissue infiltration, and undetermined margins [83,87,88]. Several studies have demonstrated that adjuvant radiotherapy post-surgery would lead to a more effective outcome of locoregional and systemic tumor control, optimizing the survival rate of cancer patients [89–91]. In severe unresectable tumors, definite radiotherapy is often prescribed. Spratt et al. (2014) reported that the five-year locoregional control rate of definite radiation comprises 57–70% of cases [92]. Another study found that the use of fast neutron radiotherapy may result in more control over the unresectable tumor than the conventional electron or photon-based therapy. However, the neutron-based method may cause more side effects and toxicity for the patients; therefore, this therapeutic intervention remains controversial [89]. Alternative therapies include carbon ion therapy, altered fractionation schedule, brachytherapy, and hyperthermia [83].

### 5.3. Chemotherapy

The application of systemic chemotherapy has been occasionally seen in severe stages of tumors with distant metastasis. A wide range of mono and polychemotherapy is used as a palliative treatment among patients for whom local therapy, such as surgery or radiation, is no longer feasible [93]. A study by Hsieh et al. (2016) reported that postoperative chemotherapy improved locoregional tumor control more than radiation [94]. However, other studies have not found any significant differences in local control and overall survival rates of chemotherapy versus radiotherapy [95,96]. Due to the absence of consistent evidence-based data, implementing chemotherapy in the treatment of salivary gland cancers adjunctively or as a palliative agent should be evaluated cautiously on a case-by-case basis.

*5.4. Other Therapeutic Interventions*

The profound comprehension of molecular behaviors in salivary gland cancers has led to the invention of other potential therapies, such as targeted therapy and hormone therapy. Tyrosine kinase inhibitors, monoclonal antibodies, and proteasome inhibitors are some of the agents used in targeted therapy [93]. Regarding hormone therapy, it has been reported that some salivary gland cancers responded well with hormonal receptors such as estrogen, progesterone, and androgen. These findings have led hormonal agents to be applied in several trial cases such as AciCC treated with tamoxifen, and both SDC and adenocarcinoma treated with antiandrogen agents [93,97,98]. Several trials in phase II are in progress to examine these new techniques; however, further investigation is needed prior to implementing this technique in cancer patients [93].

*5.5. Relative Problems of SGN Therapy*

There are risks of morbidity in all therapeutic interventions of SGNs. First of all, the complications after surgical therapy may include total or partial nerve damage, facial numbness, loss of lingual sensation, sialoceles, and salivary fistula [99]. In some cases, patients experience Frey's syndrome or gustatory sweating, which is sweating in the facial area while chewing. These complications usually take months to heal; however, in rare cases, they can be permanent [99]. Regarding the application of radiotherapy in treating SGNs, this method would also leave multiple complications to the patients. The most commonly observed consequences are dry mouth (xerostomia) and salivary gland hypofunction [100]. Multiple strategies have been proposed to lessen these manifestations and improve the patient's quality of life such as radioprotectors, preservation of salivary stem cells, or acupuncture [100]. Other surrounding structures may also be damaged by the radiotherapy, which would cause other morbidities for the cancer patients such as pharyngitis, dysphagia, dysgeusia, and trismus. These issues are usually short-term and will disappear over time [100]. However, mandibular osteoradionecrosis is a lifetime sequelae, which is often induced by prolonged and severe doses of radiation, poor dental health, post-treatment extraction, and oral trauma [101–103]. Other interventions such as chemotherapy, targeted therapy and hormonal therapy have often been used for treating recurrent and metastatic SGNs. The application of these systemic interventions is meant to relieve the cancerous-related symptoms and slow down the disease progression rate; however, there is still insufficient documentation on whether these managements could minimize the mortality rate [104]. While chemotherapy has been well-documented to result in numerous side effects to the patients such as hair loss, nausea, diarrhea, easy bleeding, and a high chance of infections [105], the results from targeted and hormonal interventions are still too restricted to report any concomitant effects [106]. In conclusion, further clinical trials with a combination of different therapies are imperative in the search for optimum treatments of SGNs.

**6. Conclusions**

Technical advances in molecular biology have helped gain deeper insight into the underlying histogenic, morphogenic, and genetic pathways responsible for various SGN. This has resulted in improved diagnostic tools and thereby more competent therapeutic modalities. However, the innately dynamic nature of salivary gland pathologies results in an everchanging classification protocol, and thus continues to challenge pathologists and clinicians. This emphasizes the need for collaborative efforts among pathologists, surgeons, medical, and radiation oncologists for personalized, case-specific treatment options with optimized prognosis.

**Author Contributions:** S.D.T., J.I., and A.H. designed and conceptualized the review. J.I., A.H., U.M.N.C., C.T.T.M., and A.W. collected the information from literature, and wrote the manuscript. C.T.T.M., A.W., B.H.N., and L.S. reviewed and edited the manuscript. P.K. worked on the illustrations and references. S.D.T. supervised the paper. All authors have read and agreed to the published version of the manuscript.

**Funding:** This research received no external funding.

**Conflicts of Interest:** The authors declare no conflict of interest.

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
