# Peer review of "An Overview on the Histogenesis and Morphogenesis of Salivary Gland Neoplasms and Evolving Diagnostic Approaches"

_cancers, doi:10.3390/cancers13153910_

Round 1

Reviewer 1 Report

General comments

While the information on histology and molecular biology seems well documented and clear, the rest (imaging, treatments) is vague, unclear or does not match up to date clinical practice and cannot be published as it is.

If it is not their field of expertise, authors should consider limiting their review to the histology and molecular biology.

Also, the paragraph on classification is not acceptable.

Specific comments

Abstract: no comments

Fig1. It is unclear where the Myoepithelial cells are found. Consider having the acini as visible by transparency if the cells are around. This does not need to be repeated in the subsequent use of the same drawing.

  1. Classification of SGN: this paragraph is too short and should provide at least one classification (most commonly accepted / most up to date).

3.1 Clinical history: should mention the accessory glands too.

3.3 to 3.7: All the imaging info is too vague and lacks precision. A table summarizing the imaging parameters for each technic that predict malignancy should be provided.

3.5 Authors state a single study (Takumi) for the efficacy of MRI derived parameters to predict malignancy. There are well known parameters and universally used in clinics that should be clearly stated here.

Table 1. For all the "Predictive", please state predictive of what?

4.3 line 339-343. "To date, there is no therapeutic intervention sufficient for this tumour, which makes it very critical [36]. ": Surgery actually cures many patients, please moderate this sentence.

Also add information on the role of post-operative radiotherapy.

5.1 Surgery: "In recurring cases, irradiation has been used as an effective treatment, since repeated surgeries in the area may induce tumor seeding and craniofacial nerve impairment [71]." Unclear. Radiation therapy is rarely efficient, and the treatment of choice is often surgery under facial nerve neuromonitoring.

Author Response

Thank you for your comments. We have significantly improved the manuscript based on comments from the 3 reviewers, and hope this revised version will satisfy you, Thank you.

Reviewer 1

General comments

While the information on histology and molecular biology seems well documented and clear, the rest (imaging, treatments) is vague, unclear or does not match up to date clinical practice and cannot be published as it is. If it is not their field of expertise, authors should consider limiting their review to the histology and molecular biology. Also, the paragraph on classification is not acceptable.

– Thank you for your comments. We have significantly improved the manuscript based on comments from the 3 reviewers, and hope this revised version will satisfy you, Thank you.

Specific comments

Abstract: no comments

Fig1. It is unclear where the Myoepithelial cells are found. Consider having the acini as visible by transparency if the cells are around. This does not need to be repeated in the subsequent use of the same drawing. – figure has been modified as suggested. The current figure illustrates the cross-sectional view of the salivary acini, enveloped by outer myoepithelial cells.

2: Classification of SGN: this paragraph is too short and should provide at least one classification (most commonly accepted / most up to date). – WHO Classification for salivary gland neoplasms 2017 has been added line (142-146), and Table 1.

3.1 Clinical history: should mention the accessory glands too.- has been added (Lines 169-170, 173), and now reads as – “SGN may be found in the parotid, submandibular, sublingual glands, accessory glands, and minor salivary glands, and most are initially identified by the swelling of these glands [22]. Furthermore, symptoms that suggest malignancy include pain, rapid tissue growth, or loss of nerve function [23, 24]. In clinical practice, it has been reported that minor SGN account for less than 25% SGN [25]…”.

3.3 to 3.7: All the imaging info is too vague and lacks precision. A table summarizing the imaging parameters for each technic that predict malignancy should be provided. – table 3 has been added elaborating imaging techniques – US, CT, MRI and PET.

3.5 Authors state a single study (Takumi) for the efficacy of MRI derived parameters to predict malignancy. There are well known parameters and universally used in clinics that should be clearly stated here. – has been modified with new references (Lines 232- 234) and now reads as – “Parameters investigating malignancy include tumour border configuration, invasion of adjacent tissues, T1- and T2- weighted signal intensity, and time-intensity curve with constant enhancement [38, 39].”.

Table 1. For all the "Predictive", please state predictive of what? – Table has been modified (current Table 4) and predictive functions of the markers has been added (Line 292 and onwards).

4.3 line 339-343. "To date, there is no therapeutic intervention sufficient for this tumour, which makes it very critical [36]. ": Surgery actually cures many patients, please moderate this sentence. – Sentence has been revised to, on lines 378-384: “Current treatment protocols involve surgery, followed by post – operative radiotherapy (PORT), which have been fairly successful [61]. PORT has shown to be an effective adjuvant to surgery and to minimize the incidence of recurrence [62]. However, studies have shown that it does not really affect the overall survival rates of patients, therefore questioning its effectiveness [63]. For more effective diagnosis and management strategies, it is necessary to understand the underlying genome alterations when studying recurrent ACC tumours [45].

Also add information on the role of post-operative radiotherapy –role of post-operative radiotherapy has been added, with new references (see above; lines 378-384).

5.1 Surgery: "In recurring cases, irradiation has been used as an effective treatment, since repeated surgeries in the area may induce tumor seeding and craniofacial nerve impairment [71]." Unclear. Radiation therapy is rarely efficient, and the treatment of choice is often surgery under facial nerve neuromonitoring. – sentence has been revised on lines 522-525 to: “Irradiation treatment exclusively has rarely been deemed as an effective treatment for SGNs. Moreover, postoperative radiotherapy is also inadvisable for benign tumors due to the associated risks of morbidity outweighing local benefits. However, in recurring cases, adjuvant radiotherapy has been proven to enhance locoregional control and reduce facial nerve damage [83].

Reviewer 2 Report

This is an interesting review about histogenesis and morphogenesis of salivary gland neplasms.

The paper is well written. However, some issues remain.

Chapter 2 must be implemented with WHO classification of salivary gland neoplasms.

In chapter 3.1, please add tumors of minor salivary glands.

In chapter 3.2, please add Milan classification of FNAC.

Author Response

We thank Reviewer 2 for her/his positive and constructive remarks on our manuscript. We have made changes to satisfy your comments. Thank you.

Reviewer 2

Comments and Suggestions for Authors

This is an interesting review about histogenesis and morphogenesis of salivary gland neoplasms.

The paper is well written. However, some issues remain.

We thank Reviewer 2 for her/his positive and constructive remarks on our manuscript. We have made changes to satisfy your comments. Thank you.

Chapter 2 must be implemented with WHO classification of salivary gland neoplasms. – WHO Classification for salivary gland neoplasms 2017 has been added (lines 142-146), and Table 1.

In chapter 3.1, please add tumors of minor salivary glands. - has been added (Lines 169-70, 173), and now reads as – “SGN may be found in the parotid, submandibular, sublingual glands, accessory glands, and minor salivary glands, and most are initially identified by the swelling of these glands [22]. Furthermore, symptoms that suggest malignancy include pain, rapid tissue growth, or loss of nerve function [23, 24]. In clinical practice, it has been reported that minor SGN account for less than 25% SGN [25]…”.

In chapter 3.2, please add Milan classification of FNAC. - has been added (Lines 182- 183), and Table 2 (Line 193).

Reviewer 3 Report

Dear Authors, thank you for your paper. I've found it interesting, well written and  readable. I suggest to modify the title like An overview on Histogenesis and Morphogenesis of Salivary Gland Neoplasms and Evolving Diagnostic Approaches more or less. Data collected on histo-morphogenesis are weel described, and pictures are excellent. However, I think that the topic is so wide to discuss all aspects of SGN in a single paper. For example, SGN of minor salivary glands should be discussed too, and the relative problems of diagnosis and therapy. The section on FNAB should better specify the diagnostic work up by cyto and histological diagnosis. Thank you again for your paper

Author Response

We wish to thank Reviewer #3 for her/his positive comments.

Reviewer 3

Comments and Suggestions for Authors

Dear Authors, thank you for your paper. I've found it interesting, well written and readable.

We wish to thank Reviewer #3 for her/his positive comments.

I suggest to modify the title like An overview on Histogenesis and Morphogenesis of Salivary Gland Neoplasms and Evolving Diagnostic Approaches more or less. – the title of the review paper has been modified to: “An Overview on the Histogenesis and Morphogenesis of Salivary Gland Neoplasms and Evolving Diagnostic Approaches”. Thank you for your recommendation.

Data collected on histo-morphogenesis are well described, and pictures are excellent. However, I think that the topic is so wide to discuss all aspects of SGN in a single paper.

For example, SGN of minor salivary glands should be discussed too, and the relative problems of diagnosis and therapy – Subheading 3 does incorporate the limitations to each diagnostic technique and subheadings 5.5 have been added (problems with therapy), with new references (lines 596 to 622).

The section on FNAB should better specify the diagnostic work up by cyto and histological diagnosis. Thank you again for your paper – has been added (Lines 182- 183), and Table 2 (Line 193).

Round 2

Reviewer 1 Report

The manuscript has been greatly improved. 

In 3.5, please list the MRI characteristics that are in favor of malignancy.

Author Response

We thank Reviewer 1 for his/her positive comments on our revised manuscript.

Thank you.
